# LDH-Co-Fe-Acetate: A New Efficient Sorbent for Azoic Dye Removal and Elaboration by Hydrolysis in Polyol, Characterization, Adsorption, and Anionic Exchange of Direct Red 2 as a Model Anionic Dye

**DOI:** 10.3390/ma13143183

**Published:** 2020-07-16

**Authors:** Nawal Drici-Setti, Paolo Lelli, Noureddine Jouini

**Affiliations:** 1Laboratoire des Sciences des Procédés et des Matériaux (LSPM), Centre National de Recherche Scientifique (CNRS), Université Sorbonne Paris Nord, LSPM-CNRS-UPR 3407, 99 Avenue Jean-Baptiste Clément, 93430 Villetaneuse, France; 2Laboratoire de Physico-Chimie des Matériaux, Département de Génie des Matériaux, Faculté de Chimie, Université des Sciences et de Technologie-Mohamed Boudiaf d’Oran (USTO-MB), M’Nouar 1505, Oran 31000, Algeria; 3Département Hygiène, Sécurité, Environnement, Institut Universitaire de Technologie, Université Sorbonne Paris Nord, 8 Place du 8 mai 1945, 93200 Saint-Denis, France; labochimie@yahoo.fr

**Keywords:** layered double hydroxide, nanomaterials, forced hydrolysis, polyol, dye removal, adsorption, anionic exchange, intercalation

## Abstract

A new, double hydroxide based on Co and Fe was elaborated on by forced hydrolysis in a polyol medium. Complementary characterization techniques show that this new phase belongs to the layered double hydroxide family (LDH) with Co^2+^ and Fe^3+^ ions located in the octahedral sites of the bucite-like structure. The acetate anions occupy interlayer space with an interlamellar distance of 12.70 Å. This large distance likely facilitates the exchange reaction. Acetates were exchanged by carbonates. The as-obtained compound Co-Fe-Ac/_Ex_ shows an interlamellar distance of 7.67 Å. The adsorption of direct red 2 by Co-Fe-Ac-LDH has been examined in order to measure the capability of this new LDH to eliminate highly toxic azoic anionic dyes from waste water and was compared with that of Co-Fe-Ac/_Ex_ and Co-Fe-CO_3_/_A_ (synthesized in an aqueous medium). The adsorption capacity was found to depend on contact time, pH, initial dye concentration, and heating temperature. Concerning CoFeAc-LDH, the dye uptake reaches a high level (588 mg/g) due to the occurrence of both adsorption processes: physisorption on the external surface and chemical sorption due to the intercalation of dye by exchange with an acetate anion. The study enables us to quantify the uptake amount of each effect in which the intercalation has the most important amount (418 mg/g).

## 1. Introduction

Layered double hydroxides (LDHs) also called hydrotalcite-like compounds are brucite-like layered materials, which consist of positively charged hydroxide layers and interlayered anions with a general formula [M^II^_1−x_M^III^_x_ (OH)_2_]^x+^·(A^n−^)_x/n_.mH_2_O, where M^II^ and M^III^ are divalent and trivalent cations, respectively. A^n−^ is an exchangeable anion located as water molecules in the interlayer space, and x is the molar ratio, M^III^/(M^II^+M^III^), which determines layer charge density [1]. These compounds have been the subject of great attention for several decades because of their use in a large number of fields ranging from the delivery of drugs to the protection against corrosion of metallurgical parts [2,3,4,5].

These performances have their origin in the very diverse chemical composition of these materials. The cationic layers accommodate a large number of metal cations while the interlayer space can accommodate a wide variety of inorganic or organic anions or even macromolecules. Besides this richness in chemical composition, LDHs exhibit interesting microstructural and structural characteristics. They present a relatively open structure due to its bi-dimensional character. Thus, the interlayer space serves as a micro-reactor where targeted anion exchange reactions can be carried out depending on the desired application (exchange reaction, exfoliation, lamination, and reconstruction.). Additionally, they can present a highly specific surface depending on the synthesis method [6,7,8,9].

In this context and thanks to their specific morphological properties and their high potential ability as ion exchangers with organic and inorganic anions, layered double hydroxides (LDHs) find their applications as sorbent agents for depolluting waste water and especially for eliminating anionic dyes, which are recognized to be a threat toward humans and ecosystems because of their high potential toxicity [10].

A review of the literature shows that the most widely used method to synthesise these materials is the coprecipitation in an aqueous medium of the hydroxides at room temperature by varying the pH of the solution. In the majority of cases, this coprecipitation is carried out in air and in the presence of sodium carbonate. In this case and whatever the nature of the starting salts, the obtained phase is a layered double hydroxide with carbonate anion intercalated between the sheets (LDH-CO_3_). This preferential intercalation is due to the great affinity of this anion for positively charged hydroxide sheets [11]. These carbonated LDHs display weak adsorption capacities. The carbonate ion is difficult to displace by an exchange reaction and, therefore, the adsorption remains limited to the surface [12].

To overcome this difficulty, one proceeds to the calcination of these materials at a moderate temperature between 400–500 °C in air. The oxides obtained are finely divided and show a memory effect. In fact, when put back into an aqueous solution, these oxides transform into LDH by intercalating the anions present in the solution. This intercalation in addition to the surface adsorption of these anions significantly improves the adsorption capacities compared to those of LDH-carbonates [13].

Another strategy consists of the synthesis of layered double hydroxides by coprecipitation in an aqueous medium where more easily exchangeable anions are intercalated such as chloride [14], nitrate [15,16], chlorate [17], dodecylsulfate [18], and sulfate [19]. To avoid the presence of CO_2_, reactions must be carried out under nitrogen, and distilled deionized water must be freshly decarbonated by vigorously boiling prior to use. 

Since the nature of the intercalated anion plays an important role in adsorption capacity, we focus in this work on LDH with acetate (Ac) as an intercalated anion. To the best of our knowledge, this anion has never been used for this purpose. Furthermore, this new layered double hydroxide containing Co^2+^ and Fe^3+^ cations in the brucite-like layers (CoFe-Ac/_p_) was produced by forced hydrolysis in a polyol medium. The advantages of this synthesis method will be discussed in light of the obtained results. The choice of the Co and Fe elements falls within the framework of our project to synthesize multifunctional LDHs based on the third transition element (Ni, Co, Fe). Like their counterparts based on Mg-Al, they present anion exchange properties, which are the basis of the work presented in this case. In addition, the presence of paramagnetic elements with several valences (Ni, Co, Fe) opens up other various application fields. These LDH are of great interest in the energy field thanks to their electrocatalytic activity for water oxidation [20]. They also serve as precursors for obtaining finely divided particles of magnetic oxides or metal alloys sought for several applications: air depollution [21], drug delivery [22], and magnetic recording [23].

To enrich and supplement this work, we investigated the anionic exchange properties of this newly synthesised LDH, along with two LDHs where carbonate anion is intercalated. The first was derived from the precursor CoFe-Ac/_p_ by exchanging acetate with carbonate anions (CoFe-Ac/_Ex_) and the second (CoFe-CO_3_/_A_) synthesised by a standard coprecipitation in an aqueous medium.

Direct red 2 was chosen as a model for the toxic azoic anionic dyes. Its adsorptive capability by sorbents is seldom reported in the literature [24,25]. The adsorption onto CoFe-Ac/_p_, CoFe-Ac/_Ex_, and CoFe-CO_3_/_A_ was studied as a function of various factors such as contact time, pH, initial dye concentration, and temperature. The results were discussed and compared to those previously reported for the adsorption of direct red 2 and similar dyes on LDH materials.

## 2. Materials and Methods

### 2.1. Materials

For all preparative procedures, Co(CH_3_COO)_2_·4H_2_O, CoCl_2_·4H_2_O, Fe(CH_3_COO)_2_, FeCl_3_, NaOH, Na_2_CO_3_, and diethylene glycol (DEG) were purchased from Acros and used without any further purification. Direct red 2 dye (purity > 99%) was provided by sigma-Aldrich and used as received. Its molecular weight is 724.73 g/mol. Figure 1 represents the estimated dimensions of the molecule obtained using Avogadro software [26].

The initial pH value of direct red 2 solution is about 6.6. A series of direct red 2 solutions of desired pH values was obtained by adjusting with dilute HCl or NaOH solutions.

#### Synthesis of LDH Samples

(a)CoFe-Ac/_p_The CoFe-Ac LDH where acetate anion is intercalated was synthesized with a molar ratio (Co/Fe) of the three following, previously described methods based on a forced hydrolysis reaction in a polyol medium [27]. Accordingly, a mixture of acetate salts dissolved in DEG with a total molar concentration of 0.1 mol/L that is heated at 130 °C under continuous stirring for 6 h. The corresponding LDH precipitated when the hydrolysis and alkalinity ratios h and b were fixed at 100 and 2, respectively, where h = nH_2_O/n (Co+Fe) and b = nNaOH/n (Co + Fe). The solid formed is separated by centrifugation. Then it is washed several times with ethanol, dried under air at 60 °C, and named CoFe-Ac/_p_. As it will be shown below by Mossbauer spectroscopy, the Fe^2+^ present in the precursor has been oxidized to Fe^3+^ in the polyol medium despite the reducing nature of this solvent. This oxidation is due to the presence of a large amount of water, which inhibits reduction, promotes oxidation, and the formation of hydroxides or oxides in addition to the easy oxidation of ferrous ions. The valence of Co^2+^ is preserved in these conditions [28,29].(b)CoFe-Ac/_Ex_The anion exchange properties of CoFe-Ac/_p_ with carbonate anions were investigated by mixing 1 g of the synthesized CoFe-Ac/_p_ LDH with 100 mL of a 2 M Na_2_CO_3_ solution. Despite the fact that the Co^2 +^ in the cationic layers appears stable with respect to oxidation (see UV-Vis-NIR analysis), the exchange has been carried out as a precaution in an inert atmosphere. After equilibrating for 24 h at room temperature, the solid was separated by centrifugation, washed several times with ethanol, dried under air at 60 °C, and then named CoFe-Ac/_Ex_.(c)CoFe-CO_3_/_A_The LDH intercalated with carbonate anions (CoFe-CO_3_) was prepared by coprecipitation in an aqueous medium [30]. An acid solution of CoCl_2_·4H_2_O and FeCl_3_, with a Co^2+^/Fe^3+^ molar ratio R = 3 and a total concentration of metallic cations of 0.75 mol/L, was added drop-by-drop to a vigorously stirred alkaline solution of NaOH (1 M) and Na_2_CO_3_ (2 M) in an inert atmosphere in order to avoid the oxidation of Co^2+^ into Co^3+^. The pH of the reaction mixture was adjusted to 10. The resulting slurry was aged at 70 °C for 24 h, separated by centrifugation, and washed extensively using distilled water until the supernatant was chloride-free, as indicated by the AgNO_3_ test. The product was dried at 60 °C under air and ground in an agate mortar. The obtained material is called CoFe-CO_3_/_A_.

### 2.2. Methods

#### 2.2.1. Characterization

The radio crystallographic characterizations and the identification of the phases were achieved using a diffractometer (INEL, Artenay, France) with Co-Kα1 radiation (λCoα1 = 1.7889 Å). The microstructure was studied by means of observations carried out in scanning electron microscopy (SEM) using LEICA STEREOSCAN 440 instrument (Cambridge, UK) and Transmission electron microscopy (TEM) performed on a JEOL-100 CX II microscope (Tokyo, Japan). The infrared spectroscopy study was carried out by transmission on a PERKIN ELMER 1750 spectrometer (Watham, MA, USA) on pressed KBr pellets with 4 cm^−1^ resolution between 400 and 4000 cm^−1^. The stoichiometry of the final product was determined by inductively coupled plasma analysis (ICP, Agilent, Santa Clara, CA, USA) at central analysis service of the national centre for scientific research (Solaize, France). The thermal stability has been specified using a Setaram TG 92-12 thermal analyser (Caluire, France). Few milligrams are introduced in an alumina crucible and submitted to thermal analysis at a heating rate of 1 °C/min under argon. A specific surface area was measured in a Micromeritics Tristar 3000 (Norcoss, GA, USA) by nitrogen adsorption N_2_ (77 K) after degassing the sample in vacuum by flowing nitrogen overnight at 100 °C.

UV-Vis-NIR (Ultra-Violet-Visible-Near Infrared) spectrum was recorded between 200 and 1800 nm with a Cary 5/Varian spectrometer (Agilent, Santa Clara, CA, USA). Polytetrafluoroethylene (PTFE) was used as a reference.

The ^57^Fe Mössbauer spectroscopic study was carried out in a transmission mode, at room temperature, using a ^57^Co/Rh γ-ray source and a conventional Mössbauer spectrometer. The spectrum was fitted by the least-squares method with a lorentzian function. The isomer shift (δ) was measured at 300 K relative α-Fe shift used as a reference. The sample (area: 3 cm^2^) is constituted of 40 mg of the studied compound dispersed in a specific resin.

#### 2.2.2. Adsorption Experiments

The adsorption experiments were carried out by a batch method in an open medium and at room temperature. A well-known mass of adsorbent (CoFe-Ac/_p_, CoFe-Ac/_Ex_, and CoFe-CO_3_/_A_ LDHs) is added in a 100 mL conical beaker to a volume of 50 mL of dye solution with a variable concentration whose temperature and pH are measured beforehand. The initial pH values and temperature were not adjusted except when the effect of these two parameters on adsorption was investigated. The adsorbent was left in contact with the dye solution for several durations. In each case, the adsorbent was then separated by centrifugation. An UV-Visible spectrophotometer (SAFAS, Monaco) was used to monitor the dye removal by measuring its remaining concentration in the solution. The measurements were made at a wavelength of λ_max_ = 500 nm, which corresponds to the maximum absorbance of the direct red 2.

The adsorption capacity q_e_ (mg/g), which represents the amount of adsorbed dye per amount of dry adsorbent, was calculated using the following equation.
(1)$$q_{e}\;=\left(C_{i}-C_{e}\right)\times \frac{V}{m}$$
where C_i_ and C_e_ are the initial and equilibrium concentrations (mg/L) of dye, respectively, and m is the mass of adsorbent (g) and V is the solution volume (L).

#### 2.2.3. Theory and Modelling

##### Kinetic Study

In order to investigate the mechanism of adsorption and to fit the kinetics experimental data, three kinetics models were used and analysed including pseudo-first-order Equation (2) [31], pseudo-second-order Equation (3) [32], and Weber’s intraparticle diffusion Equation (4) [33].
(2)$$\mathrm{Log}\;\left(q_{e}-q_{t}\right)=\;{\mathrm{Logq}}_{e}-\frac{k_{1}\times t}{2.303}$$
(3)$$\frac{t}{q_{t}}=\frac{1}{k_{2}\times q_{e}{}^{2}}+\left(\frac{1}{q_{e}}\right)\times t$$
(4)$$q_{t}=k_{p}\times t^{1/2}+C$$ where q_e_ and q_t_ are, respectively, the amount of dye adsorbed (mg/g) at equilibrium and at any time t, k_1_ is the rate constant of pseudo-first-order adsorption (min^−1^), values of k_1_ and q_e_ are determined from the plot of Log (q_e_−q_t_) = f(t), k_2_ is the rate constant of pseudo-second-order adsorption (g/mg.min). The equilibrium adsorption capacity (q_e_) and the pseudo-second-order constant k_2_ are determined from the slope and intercept of plot of t/q_t_ = f (t), k_p_ (mg·g^−1^mn^−0.5^) is the intraparticle diffusion rate constant, and C (mg·g^−1^) represents the effect of boundary layer thickness, k_p_ and C can be evaluated, respectively, from the slope and the intercept of the linear plot q_t_ = f (t^1/2^).

##### Isotherm Study

To investigate the nature of the interaction of direct red 2 anions and the LDHs adsorbents, two models were selected and used in order to match the experimental data, namely Langmuir and Freundlich isotherms. (a)Langmuir IsothermIn the Langmuir isotherm, it is assumed that the maximum adsorption is limited to a monolayer of molecules distributed homogeneously over the entire surface and without interactions between them [34]. It is given by the following linear equation.
(5)$$\frac{C_{e}}{q_{e}}=\frac{1}{Q_{\max}\times K_{L}}+\frac{C_{e}}{Q_{\max}}\;$$
where $K_{L}$ is the equilibrium adsorption coefficient (L mg^−1^), Q_max_ is the maximum adsorption capacity (mg/g), C_e_ is the equilibrium concentration (mg L^−1^), and q_e_ is the adsorbed amount at equilibrium (mg/g). $K_{L}$ and Q_max_ values were calculated from the slope and intercept of the plot of C_e_/q_e_ = f (C_e_).(b)Freundlich IsothermThe Freundlich model is based on an empirical equation, which considers that the sorption occurred on a surface where the active sites have heterogeneous energetic distribution. Additionally, it supposes multilayer adsorption with interactions between the adsorbed molecules [34,35]. It is represented by the following linear equation.
(6)$$\ln\;q_{e}=\;\ln\;K_{f}+\frac{1}{n}\times \;\ln\;C_{e}$$$C_{e}$ is the equilibrium concentration (mg·L^−1^), $q_{e}$ is the adsorbed amount at equilibrium (mg·g^−1^), and $K_{f}$ and 1/n are the Freundlich constants. The constant n is related to the energy and the intensity of adsorption and $K_{f}$ indicates the adsorption capacity (mg g^−1^). $K_{f}$ and 1/n values were inferred from the slope and intercept of the plot of $\mathrm{Ln}\;q_{e}$= f ($\ln\;C_{e}$).

##### Thermodynamic Parameters

To better understand the temperature effect on the adsorption, the thermodynamic parameters such as Gibbs-free energy change ΔG°, standard enthalpy ΔH°, and standard entropy ΔS° were studied. They were obtained from experiments at various temperatures using the following equations [36].
(7)$$\Delta G{}^{\circ}=-R\times \;T\times \;{\mathrm{lnk}}_{L}$$
(8)$${\mathrm{Lnk}}_{L}=\left(-\frac{\Delta H{}^{\circ}}{R}\right)\times \frac{1}{T}+\left(\frac{\Delta S{}^{\circ}}{R}\right)$$
(9)ΔG°= ΔH°−T×ΔS°
where $k_{L}$ is the equilibrium adsorption constant, R is the molar gas constant, and T is the absolute temperature. ΔH° and ΔS° thermodynamic parameters were calculated from the values of the slopes and the intercepts of Van’t Hoff plots of ln$k_{L}$= f (1/T).

## 3. Results

### 3.1. Characterization of Adsorbents

#### 3.1.1. X-ray Diffraction 

The X-ray diffraction patterns of all the as-prepared samples are displayed in Figure 2. They exhibit the typical signature of a crystalline layered double hydroxide belonging to the space group R-3m (JCPDS card 25-0521) [37], and display the principal characteristic reflections of LDHs by the presence of three sharp symmetrical peaks (00l) at low 2θ angle and a weak asymmetrical peaks (hk0) at high 2θ angle, which indicates a turbo-static disorder of the layers’ stacking [38]. 

The acetate anion was successfully intercalated into the interlayer gallery of CoFe-Ac/_p_ (Figure 2a), which gave an interlayer spacing of 12.70 Å. This value is very close to that of NiFe-Ac [39]. 

According to Figure 2b, it is clear that, after anion exchange by carbonate anions, the material preserved its lamellar structure and it showed the disappearance of the peak due to the acetate anions (d_003_ = 12.70 Å) and the appearance of another peak at high 2θ value attributable to the presence of intercalated carbonate anions (d_003_ = 7.67 Å). This value is in keeping with the results for the reference phase CoFe-CO_3_/_A_ (d_003_ = 7.57 Å) (Figure 2c) and those reported in the literature [1].

According to the lattice parameters of CoFe-Ac/_p_, CoFe-Ac/_Ex_, and CoFe-CO_3_/_A_ LDHs summarized in Table 1, we can say that the interlayer distance depends mainly only on the nature of the inserted anion without any change in the value of the parameter a = 2d_110_, which depends on the size of the metal cations. 

#### 3.1.2. Morphology

The samples CoFe-Ac/p and CoFe-Ac/Ex have similar morphologies. They appear formed of an aggregation of very fine platelets Appendix A. TEM images reveal that these platelets approach a hexagonal shape characteristic of LDH compounds and have a diameter around 50 nm. The sample CoFe-CO_3_/_A_ prepared by coprecipitation in an aqueous medium has a different morphology, which is also observed in the case of LDH (Appendix A) [40]. The particles are in rounded form with a micrometric size. Aggregations of these particles are also observed. TEM observations show that these particles are microporous and made up of fine particles in the nanometer size.

#### 3.1.3. Spectroscopy Studies

The infrared spectra for all synthesized compounds (Figure 3) show the characteristic absorption bands of layered double hydroxide compounds [41]. Particularly, the large band located at high frequency (3400–3500 cm^−1^) can be assigned to the OH stretching of water molecules and hydroxyl groups of the brucitic layers. The absorption band around 1640 cm^−1^ is assigned to δH_2_O vibration of the water molecules and bands at a lower wave level (ν < 800 cm^−1^) are due to vibrations implying M-O, M-O-M, and O-M-O bonds in the layer. 

As shown in Figure 3a, the intercalation of acetate anions in CoFe-Ac/_p_ LDH is confirmed by the presence of two large and intense bands in 1600–1000 cm^−1^ domain, assigned to ν_as_(-COO-) = 1561 cm^−1^ and ν_s_ (-COO-) = 1410 cm^−1^ respectively. Additionally, it is confirmed by the band with low intensity appearing at 1344 cm^−1^ that is ascribed to the vibration of the CH_3_ group (δCH_3_) of this anion. The overall spectrum in the range 1990–1300 cm^−1^ is very similar to that reported for the layered hydroxy-acetate Ni_1−x_Zn_2x_(OH)_2_(CH_3_COO)_2x_·nH_2_O [42]. The value of Δν = (νas−νs) was in the range 175 ≤ Δν ≤ 151 cm^−1^, which indicates that the acetate anions are intercalated as free species in between the layers [43]. A careful examination of the CoFe-Ac/_p_ LDH spectrum (Figure 3a) shows that the band close to 1360 cm^−1^ is absent, which suggests that the carbonate anion is not intercalated even if the contamination by this anion coming from air cannot be excluded.

In the same way, the comparison of the infrared spectrum for CoFe-Ac/_Ex_ (Figure 3b) with the spectra for CoFe-Ac/_p_ (Figure 3a) and CoFe-CO_3_/_A_ LDHs (Figure 3c) shows the complete exchange of the acetate by the carbonate anions, which confirms the results from the XRD. The two characteristic bands of the acetate ion disappear, leaving two new bands at 1494 cm^−1^ and 1363 cm^−1^ characteristic of the carbonate ion [42].

Lastly, for both CoFe-Ac/_p_ and CoFe-Ac/_Ex_ compounds, the presence of absorption bands at low intensity located at approximately 2930 and 2854 cm^−1^ along with two weak absorption bands are located between 1300 and 1000 cm^−1^. They are attributed to the presence of the DEG solvent adsorbed on the surface of LDHs [39].

The UV-Vis-NIR spectrum of CoFe-Ac/_p_ is shown in Appendix A. We note the presence of the absorption bands located at around 1200 to 500 nm, which are attributed to a high spin transition of octahedral coordinated Co^2+^ [44]:ν_1_ (Co^2+^_Oh_):^4^T_1g_(F) → ^4^T_2g_(F)
ν_2_ (Co^2+^_Oh_):^4^T_1g_(F) → ^4^A_2g_(F)
and ν_3_ (Co^2+^_Oh_):^4^T_1g_(F) → ^4^T_1g_(P)

The absorption band at around 700 nm is absent, which corresponds to Co^3+^ ions in octahedral sites. Similarly, the results show the absence of Co^2+^ at tetrahedral sites since the characteristic bands located around 1567 nm for d-d transitions are absent:ν_2_ (Co^2+^_Td_): ^4^A_2_(F) → ^4^T_1_(F), ν_2_ (Co^2+^_Td_): ^4^A_2_(F) → ^4^T_1_(F)
and, at around 592 and 641 nm, for the transition [45]: ν_3_ (Co^2+^_Td_: ^4^A_2_(F)) → ^4^T_1_(P)

The UV-Vis-NIR study was conducted six months after the synthesis of the compound was kept for this period in air. It clearly shows that the compound is stable with respect to oxidation, as shown by the absence of Co^3+^.

Appendix A shows the Mössbauer spectrum for the CoFe-Ac/_p_ sample at room temperature in order to probe the local magnetic environment around the Fe sites and to determine the oxidation state of iron in the LDH matrix. The spectrum shows only one doublet with an isomer shift δ(Fe^3+^) = 0.339 mm/s and quadrupole splitting Δ(Fe^3+^) = 0.44 mm/s, which indicates an Fe^3+^ nature of the Fe atom, and no evidence of the Fe^2+^ signal was found. The small value of quadrupole splitting (Δ) corresponds to high spin Fe^3+^ ions in octahedral sites [46].

#### 3.1.4. Thermal Analysis

The TG/DTA (Thermogravimetry/Differential Thermal Analysis) for CoFe-Ac/_p_ LDH is presented in Figure 4. The TG diagram is characteristic of LDH thermal behaviour showing three weight losses [47]. Upon heating, we observe a weight loss of 16% due to the departure of adsorbed and interlayer water molecules. This departure is characterized by two endothermic peaks at 97 °C and 168 °C. The de-hydroxylation of the brucite-like layers is characterized by the endothermic peak at 262 °C and corresponds to the second weight loss of 11.5%. Lastly, the third step occurred in a large range of temperature (342–700 °C). The corresponding total loss (28.5%) actually corresponds to the loss of acetate ions (10%) along with that adsorbed DEG molecules (7%) and also corresponds to the release of oxygen (11%) due to the reduction of cobalt ions. X-ray diffraction analysis shows that the residue is a mixture of iron oxide and metallic cobalt.

#### 3.1.5. Chemical Analysis

Table 2 summarizes the results of elemental chemical analysis of the CoFe-Ac/_p_ compound and its corresponding formula. The calculated Co^II^/Fe^III^ ratio aligns with the experimental data. The observed carbon content is slightly higher than that corresponding to the acetate anions. This confirms the results of the infrared study showing the presence of DEG molecules likely adsorbed on the surface of the particles. This appears as a common feature of inorganic compounds prepared in a polyol medium [48]. Thus, we have taken into account the amount of adsorbed polyol in the chemical formula in order to calculate the theoretical weight loss, which matches that observed by TG analysis (Table 2).

#### 3.1.6. Surface Area Measurements 

The nitrogen adsorption-desorption isotherms for the CoFe-Ac/_p_ and CoFe-Ac/ex phases are shown in Appendix A and the results of the surface area measurement are included in Appendix A. The two phases present almost similar textural characteristics. Their surface areas are very close at 48 and 50 m^2^/g, respectively, and their isotherms match those recorded for a layered double hydroxide type [49]. The shape of the isotherms is type IV, which indicates a predominant mesoporous character with pores measuring between 2 and 50 nm [50]. 

### 3.2. Adsorption Study

#### 3.2.1. Effect of Contact Time

The effect of contact time on the removal of direct red 2 by all LDHs adsorbents illustrated in Figure 5 was studied at room temperature at a pH equal to 6.6 and with initial dye concentration of 1 g/L. It is clear that the fastest kinetics is observed for CoFe-Ac/_p_ LDH where the shape of the curve indicates fast diffusion. In this case, the highest quantity of dye (94%) is already fixed at approximately 5 min, which is followed by a slower stage until reaching the total removal. This corresponds to an equilibrium time of 15 min. For the two other materials, the adsorption was slower and their equilibrium time was reached at about 180 and 240 min with regard to CoFe-Ac/_Ex_ and CoFe-CO_3_/_A_, respectively.

#### 3.2.2. Kinetic Modelling

The calculated kinetic parameters for direct red 2 adsorbed by CoFe-Ac/_p_, CoFe-Ac/_Ex_, and CoFe-CO_3_/_A_ are reported in Table 3. Experimental results fit the first order kinetic model with regression coefficient values of 0.9599, 0.9652, and 0.9128 for CoFe-Ac/_p_, CoFe-Ac/_Ex_, and CoFe-CO_3_/_A_, respectively. Moreover, large differences between experimental and calculated values of the equilibrium adsorption capacities are observed. On the contrary, the second-order-kinetic model curves (Appendix A) show much better correlation coefficients (value higher than 0.99) in all cases and the calculated q_e_ is much closer to the experimental values. The second-order-kinetic model seems to be the more appropriate one to describe the adsorption of anionic dye on LDHs [51].

As shown in Table 3, k_2_ (pseudo-second-order constant) is inversely related to the equilibrium time. A high value of k_2_ implies a shorter kinetic for the same concentration and that the limiting adsorption step is a chemical interaction adsorbate-adsorbent [51,52]. This is the case of adsorption of direct red 2 on CoFe-Ac/_p_. None of the two kinetic models could identify the diffusion mechanism. Thus, the intra-particle diffusion model was further investigated.

Figure 6 shows the variation of q_t_ versus t^1/2^. Even if the values R^2^ vary between 0.8108 and 0.9768 for all systems and remain lower than those found for the pseudo-second order model (Table 3), this model brings interesting insight into the adsorption mechanism occurring.

The curves are composed of two linear segments indicating the presence of two steps and confirming that the intraparticle diffusion is not the only process implied for these systems since a simple straight line would be observed if this was the case. 

According to several studies, the first segment corresponds to the instantaneous step that controls the diffusion at the external surface. When the external surface saturates, the dye diffuses inside the particle and allows the ion exchange to occur in between the layers as will be observed in the discussion section. This progressive intra-particle diffusion process is presented by the second segment [19,52].

From the same curves, it is clear that the lines do not pass through the origin. That can be due to the difference in the rate of mass transfer during the initial and final stages of adsorption. It also states that the intraparticle diffusion is not the only stage limiting adsorption and that a surface adsorption can occur simultaneously [53].

The k_p_ values indicate (Table 3) that the intraparticle diffusion is affected by the type of material used for the adsorption. We notice that the diffusion in the case of CoFe-Ac/_p_ is most significant (k_p_ = 5.4882 mg/g·mn^1/2^). That is likely due to the appearance of a significant porous surface for this material along with the ease of access to the interlamellar space, as will be shown in the discussion section. The high values of C can be due to a great contribution of surface adsorption on the phenomenon of the intraparticle diffusion [54].

#### 3.2.3. Effect of pH

The initial pH of the dye solution is considered an important parameter, which controls the adsorption at water-adsorbent interfaces, it affects the surface charge of adsorbent as well as the degree of ionization of pollutants. Therefore, the adsorption of direct red 2 on the different LDHs adsorbents was examined at different concentrations and different pH values ranging from 7 to 10 at 20 °C. As shown in Figure 7, it was observed that the adsorption capacity of direct red 2 on CoFe-Ac/_p_ is practically constant and it was nearly independent of pH in the pH range of 7–9 (Figure 7a). Similar results were obtained for the adsorption of the indigo carmine dye by calcined hydrotalcite MgAlCO_3_ [55].

However, a remarkable diminution in dye adsorption on all materials occurred when the pH was greater than 9. This diminution was greater at a high concentration of dye solutions. 

For CoFe-Ac /_Ex_ and CoFe-CO_3_/_A_ LDHs (Figure 7b,c), a decrease in the adsorption capacity is observed when the pH increases from 7 to 10. This may be due to the presence of competitive adsorption of OH^−^ and CO_3_^2−^ with coloured ions. These anions have a very great affinity for LDHs [56]. Similar results were found for adsorption of fluoride on calcined hydrotalcite [57] and for adsorption of selenite on calcined, layered double hydroxide MgFe-CO_3_ [51]. 

#### 3.2.4. Effect of Temperature and Thermodynamic Study

The temperature is a very significant parameter for the adsorption process. For this reason, the adsorption of direct red 2 on CoFe-Ac/_p_, CoFe-Ac/_Ex_, and CoFe-CO_3_/_A_ LDHs was studied by carrying out a series of isotherms at 283, 293, and 323 K.

As shown in Figure 8, it is clear that sorption capacities of dye onto all adsorbents increased with an increasing temperature from 283 to 323 K at a high concentration of solution dye, which implies that adsorption of direct red 2 onto all materials is endothermic in nature.

This result may be attributed to the enhanced mobility of direct red 2 ions at high temperature due to greater vibrational energies of the molecules, which facilitates the penetration of dye into the internal structure of LDHs and increases its diffusion in the pores [58]. These results fit well with those described for treating methyl orange by calcined, layered double hydroxide ZnAl-CO_3_ [59].

ΔG°, ΔH°, and ΔS° values are reported in Table 4. For all adsorbents, the positive values of ΔH° (ΔH° > 0) confirm that the adsorption process is endothermic [60]. The decrease in the negative ΔG° values with an increase in temperature show the spontaneous nature of adsorption, and indicates that the adsorption process becomes more favourable at a higher temperature [61].

The positive values of standard entropy ΔS° reflect the affinity of all adsorbent LDHs for direct red 2 dye in an aqueous medium and suggest that some structural changes occur on the adsorbent with the increase of randomness at the solid/liquid interface in the adsorption system [62].

#### 3.2.5. Adsorption Isotherms

Adsorption data are often presented as an adsorption isotherm, which is important for understanding the interactions between adsorbent and adsorbate. Figure 9 shows the adsorption isotherms of direct red 2 on the three studied LDHs. They all display the typical L shape Sub-group 2 (Langmuir monolayer) according to Giles’s classification [63], which indicates a great affinity between the adsorbate and the adsorbent and corresponds to the formation of a monolayer of dye.

According to the same figure, we can also observe the decrease in adsorption capacities (Q_max_) following the series: CoFe-Ac/_p_ (≈588 mg/g) > CoFe-Ac/_Ex_ (≈170 mg/g) > CoFe-CO_3_/_A_ (≈127 mg/g).

According to the curves of the tested models (Langmuir, Freundlich), Appendix A and, following the higher correlation coefficients R^2^ values close to 1 (Table 5), we can note that the Langmuir model is the best to describe the phenomenon of adsorption. The Toth model confirms that the Langmuir isotherm better describes this adsorption [64] (Appendix A).

The Langmuir parameters Q_max_ and K_L_ were calculated and listed in Table 5. As shown, the K_L_ parameter reflects the affinity of LDHs for direct red 2 [65] and the calculated maximum adsorption capacities are very close to those obtained in experiments. However, the best adsorption capacity is observed for the CoFe-Ac/_p_ (Q_max_ ≈ 588 mg/g), which corresponds to an uptake capacity of 3.4 times greater than that of CoFe-Ac/_Ex_ and 4.6 times greater than that of CoFe-CO_3_/_A_. CoFe-Ac/_p_ LDH prepared here in polyol medium presents high adsorption capacity of direct red 2 than CTAB-bentonite (≈109.89 mg/g) [24] and calcined hydrotalcite (≈417mg/g) [25]. Furthermore, this new adsorbent shows removal efficiency of 100% until 1.5 g/L of dye solution.

### 3.3. X-ray and IR Characterizations of the LDHs after Adsorption 

Infrared spectra clearly confirm the adsorption of direct Red 2 on the three LDHs: CoFe-Ac/_Ex_, CoFe-Ac/_p_ (Figure 10), and CoFe-CO_3_/_A_ (Appendix A). The main characteristic bands of this dye are present. The absorption bands located between 1250 and 1000 cm^−1^ are assigned for the sulfonic SO_3_^−^ vibrations group. The absorption band at approximately 1600 cm^−1^ is attributed to the absorption of C-C aromatic stretch and the band at around 1500 cm^−1^ is ascribed to the azoic group [14]. 

However, the IR spectra of CoFe-CO_3_/_A_ and CoFe-Ac/_Ex_ after adsorption still present the same asymmetrical shape in 1600–1300 cm^−1^ region as that before adsorption. In this region, we notice the intense band at 1364 cm^−1^ showing the predominance of the carbonate ion in the interlamellar space. Conversely, the spectrum of CoFe-Ac/_p_ after adsorption has a more symmetrical shape in the same region. We observe the disappearance of the characteristic bands of acetate ion and the appearance of four bands in which two are intense and coincide with those of the direct red 2 anion, which shows that the acetate has been exchanged by this anion. However, the concomitant intercalation of the carbonate ion coming from air cannot be excluded since spectra of the two species overlap in the 1600–1300 cm^−1^ domain.

X-ray diffraction analysis sheds additional light. It shows that CoFe-CO_3_/_A_ (Appendix A) and CoFe-Ac/_Ex_ (Figure 11A) maintain, before and after adsorption, the same interlayer distance revealing that the carbonate has not been exchanged by the direct red 2 anion and that adsorption has occurred only on the surface of the particle.

Conversely, in the case of CoFe-Ac/_p_ (Figure 11B), the characteristic interlamellar distance of the acetate ion (12.70 Å) disappears after adsorption, which results in two new distances (23.7 Å and 8.28 Å). This reveals that the direct red 2 anion was intercalated in the interlayer space. 

## 4. Discussion: Mechanism of Direct Red 2 Removal and Comparison with Previous Works

Table 6 compares the results obtained during this work with the adsorption performance of azoic dyes belonging to the same family (with at least two sulfonate groups) by LDHs prepared via the common synthetic routes discussed in the introduction.

As can be seen, CoFe-CO_3_/_A_ and CoFe-Ac/_Ex_ behave similarly to LDHs prepared by coprecipitation in an air atmosphere. Their final interlamellar distances remain unchanged after adsorption and their removal capacities are low (128 and 175 mg/g). This is due to the great affinity of the carbonate anion with the positively charged LDH layers. Thus, the anionic exchange between the carbonate and the anionic dye is energetically disadvantaged and, exclusively, surface adsorption takes place (physisorption). This physisorption is confirmed by the small positive values of ΔH° (ΔH° < 40 kJ/mol) obtained for these two compounds (Table 4) and by the low k_2_ constant of the second order kinetic model (Table 3).

As shown in Table 6, LDHs with carbonate anion heated at about 500 °C present significantly higher adsorption capacities due to the high surface properties and to the memory effect resulting after calcination. In this case, besides physisorption on the external surface, the intercalation of the anionic dye appears to be the main sorption process during the reconstruction of the layer structure. The intercalation may lead to anionic dye standing perpendicularly to the lamellar sheets and giving a high basal spacing. The anionic dye may also be intercalated in a flat position, which results in a limited interlamellar distance [69]. Even if the calcination-reconstruction process clearly improves the LDH adsorption capacities, the intercalation of the anions remains below the theoretical value allowed by the M^2+^/M^3+^ ratio. The charge compensation is fulfilled by the simultaneous intercalation of the carbonate anion present in the effluent as clearly shown by IR studies. Among the examples cited in Table 6, only the intercalation of Congo Red into calcined Zn-Al-CO_3_ reaches the theoretical value (1540 mg/g). The performance of LDHs also prepared by coprecipitation and containing other intercalated anions depends on the size of these anions. LDH intercalated with small anions: nitrate, chloride, and sulfate have interlamellar distances very close to the LDH-CO_3_ distance. Their adsorption performance is variable as shown in Table 6. For sulfate and nitrate, the adsorption capacity is medium or even low. Significantly larger anions such as dodecylsulfate (SDS) lead to the formation of LDH with a significant interlamellar distance (25.5 Å). This weakens cohesion interlayers and, therefore, facilitates the exchange. Such conditions are at the origin of the high adsorption capacity observed in this case (707.76 mg/g).

The sorbent proposed in the present work (LDH-CoFe-Ac/_p_) is based on a new approach. This material was prepared by forced hydrolysis in a polyol medium. This route, described for the first time for the preparation of the Layered Hydroxide Salts (LHS) [70], has been extended to the synthesis of LDH based on the 3D transition elements [27,39,71]. This synthesis method presents two main advantages: (i) the reaction is conducted without atmosphere control since polyol avoids the carbonate anions’ contamination and (ii) the pH is mainly controlled by the amount of acetate anion present in the precursor salts [70]. In this new LDH, the acetate anion was chosen as the intercalate anion. This leads to a relatively large inter-layer distance (12.70 Å). As dodecylsulphate, this likely weakens the cohesion of sheets and, thus, will favor the exchange reaction. Such factors enhance the adsorption capacity leading to the high adsorption efficiency observed for this compound (588 mg/g). The high ΔH° value obtained ΔH° > 40 kJ/mole means that the adsorption is characterized by a physisorption effect followed by a chemical effect (intercalation reaction). This is in keeping with results reported on the sorption of EB and similar dyes on ZnAl-Cl [14] and the sorption of direct red 2 on calcined hydrotalcite [25]. The occurrence of both physisorption and chemical sorption via intercalation is also confirmed by the high constant K_2_ of the second order-kinetic model (Table 3).

X-ray diffraction analyses clearly confirm these results. They show that the adsorption processes are mainly controlled by the dye concentration. When the concentration is low (100 mg/L), only physisorption occurred (adsorption on the external surface). No change in basal spacing (12.70 Å) was observed. XRD patterns remain identical before and after adsorption (Figure 11B(a,b)). At a higher concentration, the chemisorption phenomenon (exchange/intercalation) begins to act in addition to the surface adsorption. The basal spacing 12.70 Å disappeared giving way to the appearance of two new distances (23.7 Å and 8.28 Å) (Figure 11B(c–f)).

These results allowed us to deduce that the intercalation of direct red 2 gave rise to the formation of two LDH phases with the same brucite-like layers but with two different interlamellar distances. The first phase corresponding to the highest interlamellar distance (23.7 Å) results from the intercalation of direct red 2 in an inclined orientation toward the brucite-like layers. Taking into account the long axis length (26 Å) (Figure 1) and the thickness of the brucite-like layer (4.8 Å), the inclination angle calculated is about 46°. Similar interlamellar distance was obtained for the adsorption of the same dye on calcined hydrotalcite [25].

The second phase is characterized by two peaks: an intense, symmetrical peak at 8.28 Å and its harmonic at 4.13 Å with dissymmetrical shape and low intensity. This lamellar phase with an interlamellar distance of 8.28 Å can result from the intercalation of direct red 2 in an almost flat position parallel to the layers, as was previously observed for RR-3FB intercalated into MgAl-SO_4_ [19] and for RR-X-3B intercalated into calcined ZnAl-CO_3_ [69]. The short axis length of direct red 2 (3.9 Å) (Figure 1) is very close to that of RR-3FB (3.7 Å) and to that of RR-X-3B (4 Å). Taking into account the thickness of the brucite-like layer, a predicted interlamellar distance of 8.70 Å would be expected. This value corresponds to what was observed. This orientation corresponds to a slight inclination (7°) of the direct red 2 long axis relative to the layers.

There exists a balance between the two phases depending on the dye concentration in the solution. Relatively low concentrations favour the lamellar phase corresponding to the direct red 2 anion intercalated almost parallel to the layers (600 mg /L, Figure 11B(c)) (Interlamellar distance 8.28 Å). Higher concentrations facilitate the organization of anions in an inclined position relative to the layers (Interlamellar distance 23.7 Å) (Figure 11B(d–f)). 

Lastly, the comparative study of both CoFe-Ac/_P_ and CoFe-Ac/_Ex_ brings an interesting insight into the relative importance of physisorption (external surface adsorption) and chemical sorption (intercalation). CoFe-Ac/_Ex_, containing carbonate as an intercalate anion, was obtained by a topotactic anion exchange reaction from the CoFe-Ac/_P_ compound. It follows that its microstructure likely remains identical (porosity and specific surface). Thus, we can suppose that the external surface adsorption for CoFe-Ac/_p_ is very close to that of CoFe-Ac/_Ex_ (i.e., 170 mg/g). This makes it possible to estimate the contribution of intercalaion around 418 mg/g showing that chemical sorption is predominant. It corresponds to 2/3 of the theoretical exchange capacity. This difference observed with respect to the theoretical intercalation rate appears to be a common characteristic of the LDHs used as adsorbents, regardless of their synthesis method (see references cited in Table 6 and Reference [17]). In the present case, the compensating interlayer anions are mostly acetate. However, the concomitant intercalation of the carbonate ion coming from air cannot be excluded since the spectra of the two species overlap in the 1600–1300 cm^−1^ domain. Such a phenomenon was also observed regarding other organic dyes such as Methyl Orange adsorbed on Ni and ZnAl-LDH where compensating anions are nitrate and/or carbonate [72] and in the case of fluorescein on ZnAl-Hydrotalcite with perchlorate as a compensating anion [17].

## 5. Conclusions

In the present study, a new CoFe-Ac/_P_ LDH was prepared by forced hydrolysis in a polyol medium, and its structural properties were confirmed by complementary characterization techniques.

The synthesised CoFe-Ac/_P_ LDH has shown anion exchange properties. Indeed, acetate interlayer species were successfully exchanged by carbonate anions with a topotactic reaction.

The adsorption capacities of dye on CoFe-Ac/_P_ were significantly affected by the initial contact time, initial dye concentration, pH, temperature, and by the structural properties (the nature of the initial interlayer anion and morphology).

In comparison with CoFe-Ac/_Ex_ and CoFe-CO_3_/_A_ LDHs, it is demonstrated that the CoFe-Ac/_P_ compound displays unique adsorption properties for direct red 2 dye. The adsorption was induced by physisorption at a low concentration of dye. High concentrations of dye favour the intercalation of dye anions via the exchange reaction with acetate anions. In addition to physisorption, this intercalation, which is considered as a chemical sorption, confers to this material an efficient uptake capacity (≈588 mg/g).

The intercalation of the azo dye anion leads to the formation of two LDH phases differing in the orientation of this anion relative to the layers. The first phase corresponds to an intercalation of the anion almost parallel to the layers. In the second phase, the azoic anions are inclined about 46° relative to the layers. Higher concentrations favour the latter orientation.

Lastly, when taking into account the obtained results, the forced hydrolysis in a polyol medium constitutes an original way and easy method to elaborate layered double hydroxides exempt of contamination by carbonates with controlled morphology, nanometric size, a more significant surface area, and, consequently, a good dispersion of particles. In addition, the acetate anion has been chosen as the intercalated specie since it leads to high interlamellar distance facilitating the exchange reaction. Altogether, these characteristics are at the origin of the efficient adsorbent capacity of this material and, accordingly, confer on it a major interest in the retention of the pollutants contained in the industrial effluents.

## Figures and Tables

**Figure 1 materials-13-03183-f001:**
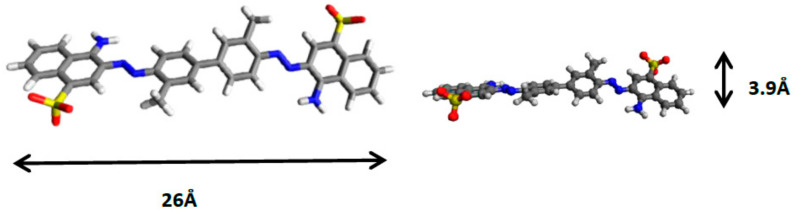
Estimated dimensions of direct red 2 anionic dye.

**Figure 2 materials-13-03183-f002:**
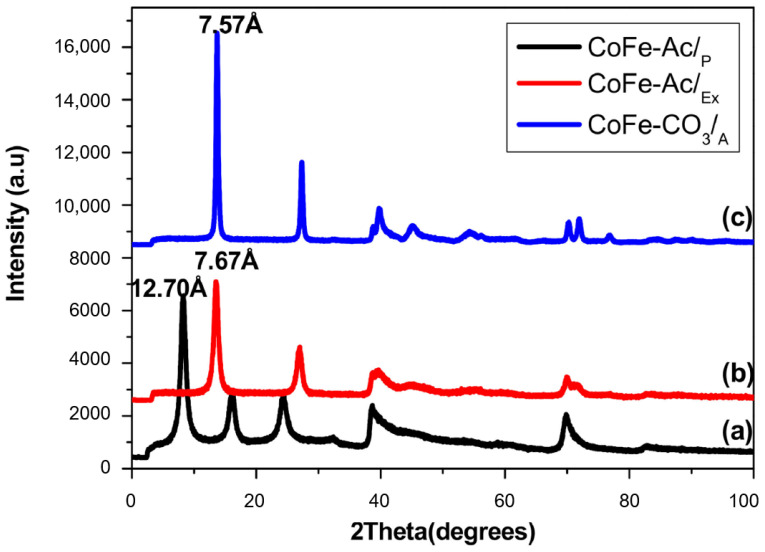
X-ray diffractogram of (**a**) CoFe-Ac/_p_, (**b**) CoFe-Ac/_Ex_, and (**c**) CoFe-CO_3_/_A_ LDHs.

**Figure 3 materials-13-03183-f003:**
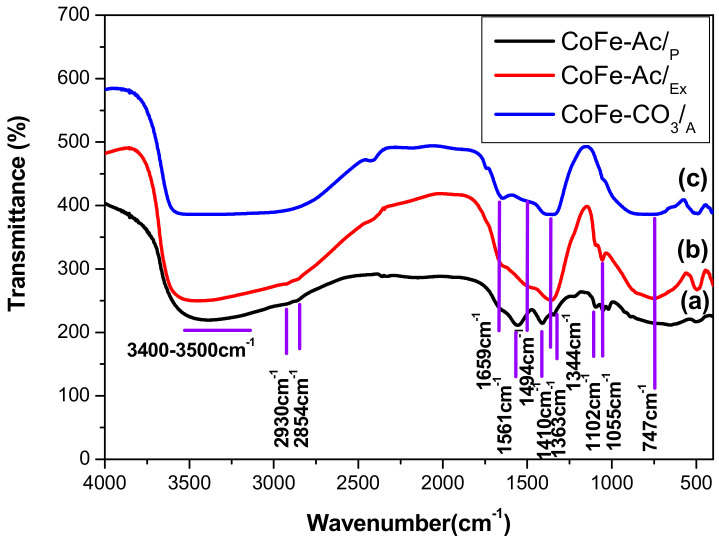
FT-IR spectra for **a** CoFe-Ac/_p_, (**b**) CoFe-Ac/_Ex_, (**c**) CoFe-CO_3_/_A_ LDHs.

**Figure 4 materials-13-03183-f004:**
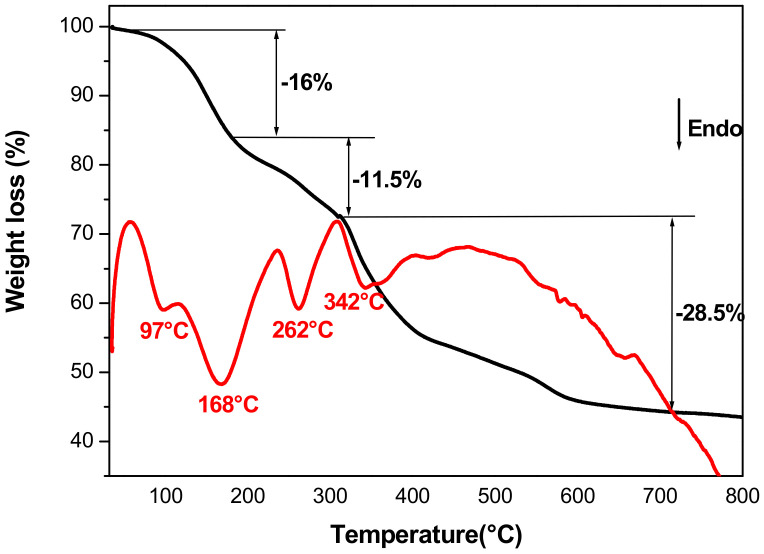
TG/DTA curves of CoFe-Ac/_p_ LDH.

**Figure 5 materials-13-03183-f005:**
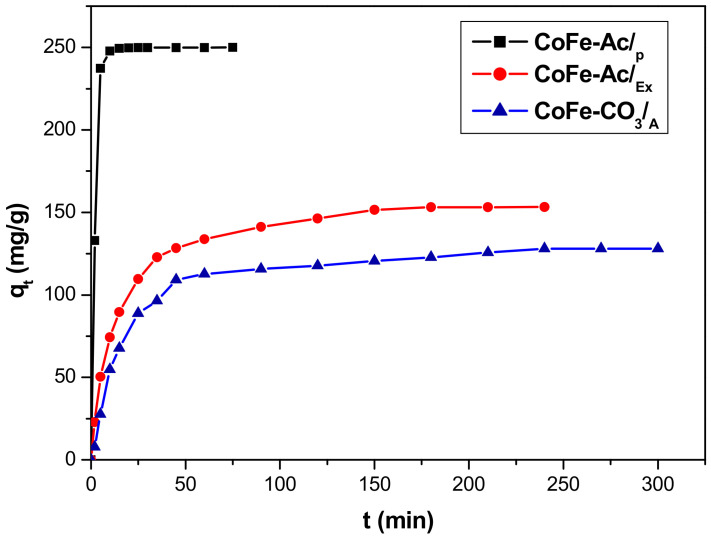
Adsorption kinetics of direct red 2 on CoFe-Ac/_p_, CoFe-Ac/_Ex_, and CoFe-CO_3_/_A_ LDHs. (C = 1 g/L, pH = 6.6, T = 20 °C).

**Figure 6 materials-13-03183-f006:**
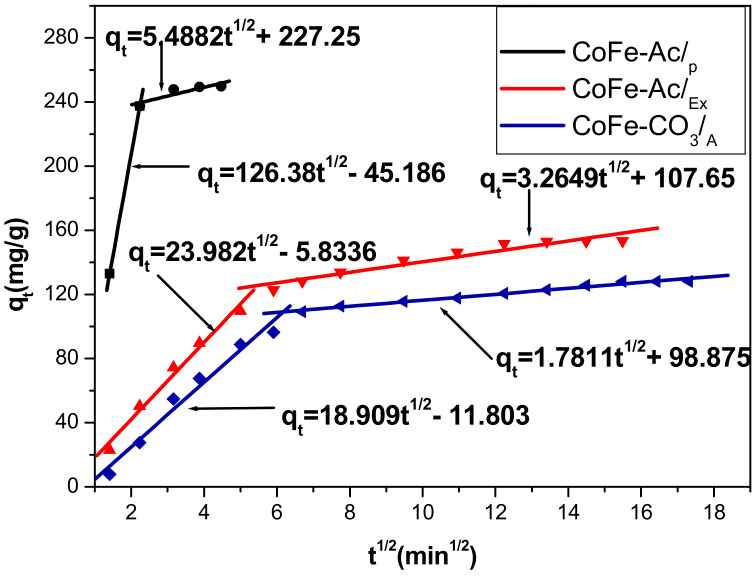
Intraparticle diffusion plot for adsorption of direct red 2 on CoFeAc/_p_, CoFe-Ac/_Ex_, and CoFe-CO_3_/_A_ LDHs.

**Figure 7 materials-13-03183-f007:**
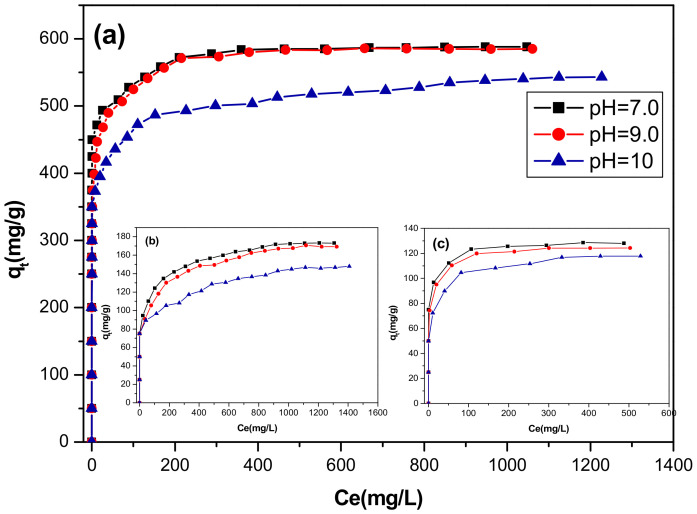
Effect of initial pH on adsorption isotherms of direct red 2 on: (**a**) CoFe-Ac/_p_**,** (**b**) CoFe-Ac/_Ex_, and (**c**) CoFe-CO_3_/_A_ LDHs.

**Figure 8 materials-13-03183-f008:**
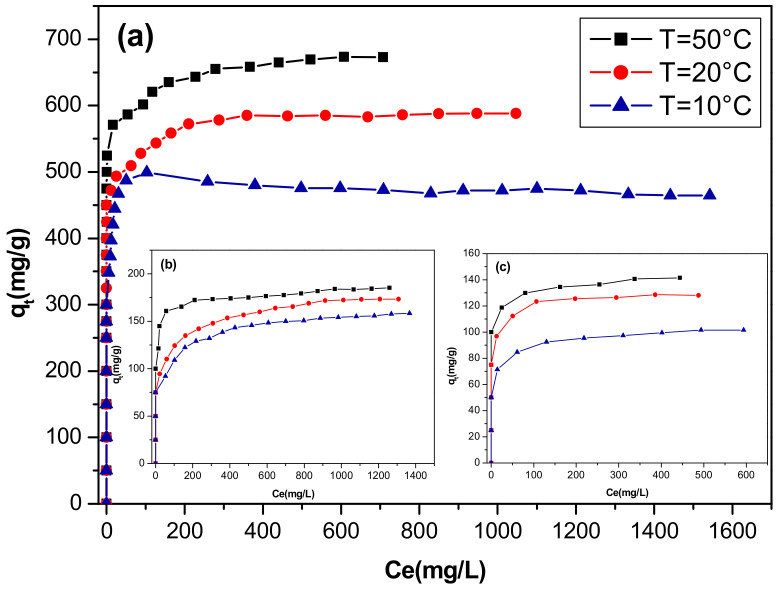
Effect of temperature on adsorption isotherms of direct red 2 on (**a**) CoFe-Ac/_p_**,** (**b**) CoFe-Ac/_Ex_, and (**c**) CoFe-CO_3_/_A_ LDHs.

**Figure 9 materials-13-03183-f009:**
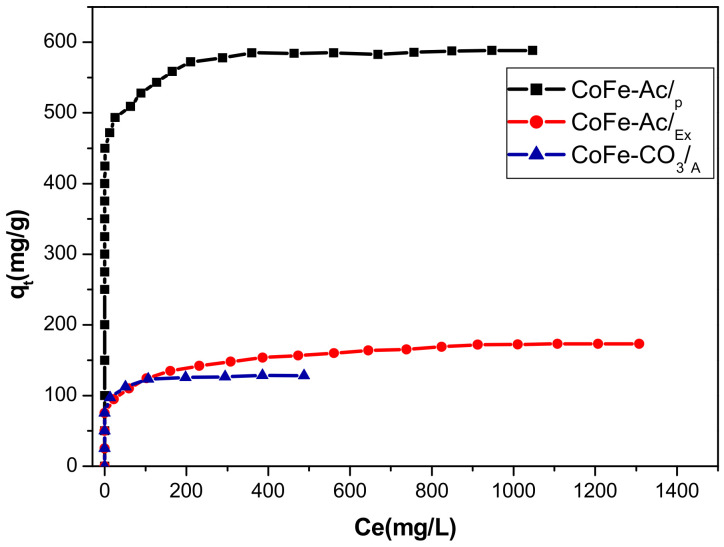
Adsorption isotherms of direct red 2 on CoFe-Ac/_p_, CoFe-Ac/_Ex_, and CoFe-CO_3_/_A_ LDHs. (pH = 6.6, T = 20 °C).

**Figure 10 materials-13-03183-f010:**
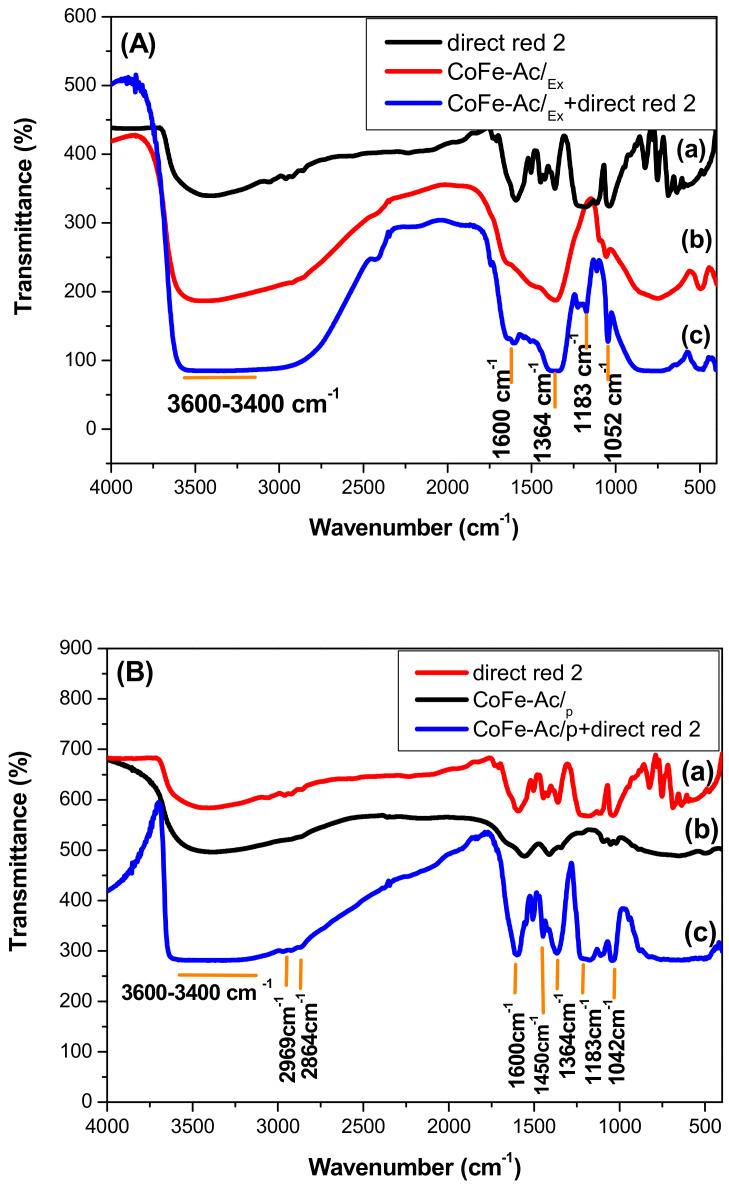
FT-IR spectra before and after adsorption of dye for (**A**) CoFe-Ac/_Ex_, (**B**) CoFe-Ac/_p_ LDHs.

**Figure 11 materials-13-03183-f011:**
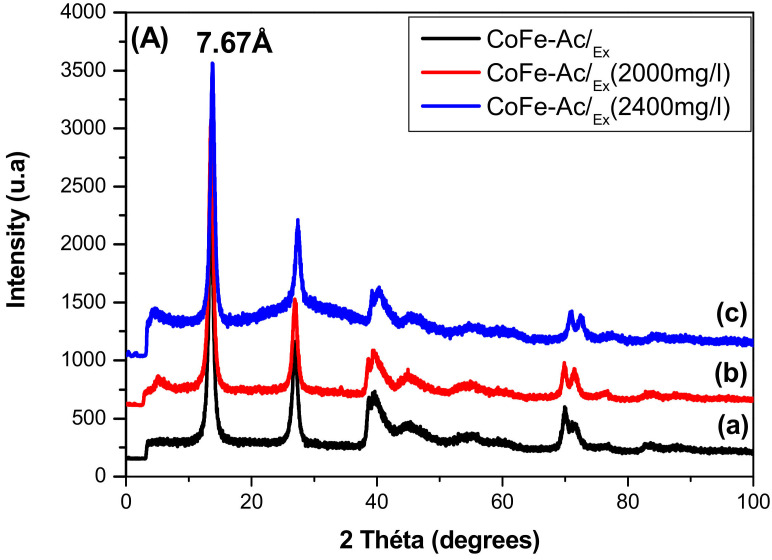
X-ray diffractogram before and after adsorption of dye for (**A**) CoFe-Ac/_Ex_, (**B**) CoFe-Ac/_p_ LDHs.

**Table 1 materials-13-03183-t001:** Evolution of the lattice parameters for CoFe-Ac/_Ex_ compared to CoFe-Ac/_p_ and CoFe-CO_3_/_A_ (reference) LDHs.

Compound	d_003_ (Å)	a Parameter (Å)
CoFe-Ac/_p_	12.70	3.11
CoFe-Ac/_Ex_	7.67	3.12
CoFe-CO_3_/_A_ (reference)	7.57	3.12

**Table 2 materials-13-03183-t002:** Elemental chemical analysis data of CoFe-Ac/_p_ LDH.

Compound	Mass Fraction (%)				Molar Ratio Co^II^/Fe^III^		X = Fe^III^/Co^II^ + Fe^III^			
	Co	Fe	C	H	Solution	Solid	Solution		Solid	
CoFe-Ac/_p_	28.68	8.96	6.80	4.34	3.0	3.0	0.25		0.25	
Chemical formula							DEG	H_2_O%	Total weight loss %	
								Exp.	Exp.	Cal.
Co_0.75_Fe_0.25_·(OH)_2_·Ac_0.25_,1.50H_2_O, 0.094DEG							0.094	16.0	56.0	54.4

**Table 3 materials-13-03183-t003:** Kinetic parameters for the adsorption of direct red 2 onto LDH samples.

Method/Adsorbent				
**Pseudo-First-Order**	**q_exp_ (mg/g)**	**k_1_ (min^−1^)**	**q_e_ (mg/g)**	**R^2^**
CoFe-Ac/_p_	250	0.494	234.54	0.9599
CoFe-Ac/_Ex_	153	0.026	101.24	0.9652
CoFe-CO_3_/_A_	128	0.016	70.24	0.9128
**Pseudo-Second-Order**	**q_e_ (mg/g)**	**k_2_·10^3^ (g/mg·min)**	**R^2^**	**-**
CoFe-Ac/_P_	250	8.42	0.9996	-
CoFe-Ac/_Ex_	161.3	0.529	0.9999	-
CoFe-CO_3_/_A_	136.98	0.426	0.9978	-
**Intraparticule Diffusion**	**k_p_(mg/g·mn^1/2^)**	**C (mg/g)**	**R^2^**	**-**
CoFe-Ac/_p_	5.4882	227.25	0.8108	-
CoFe-Ac/_Ex_	3.2649	107.65	0.9271	-
CoFe-CO_3_/_A_	1.7811	98.875	0.9768	-

**Table 4 materials-13-03183-t004:** Thermodynamic parameters for the adsorption of direct red 2 onto LDH samples.

Compound	ΔH° (kJ/mol)	ΔS° (J/mol·K)	ΔG° (kJ/mol)		
			283K	293K	323K
CoFe-Ac/_P_	41.31	202.54	−16.00	−19.04	−24.11
CoFe-Ac/_Ex_	8.75	78.89	−13.57	−14.76	−16.73
CoFe-CO_3_/_A_	11.48	84.88	−12.54	−13.81	−15.93

**Table 5 materials-13-03183-t005:** Isotherm parameters for the adsorption of direct red 2 onto LDH samples.

Method/Adsorbent				
Langmuir Isothem	Q_max_ (mg/g)	K_L_ (l/mg)	R^2^	Q_max_ (exp) (mg/g)
CoFe-Ac/_p_	588.23	0.404	0.9999	588
CoFe-Ac/_Ex_	175.44	0.034	0.9998	170
CoFe-CO_3_/_A_	128.205	0.426	0.9998	127
**Freundlich Isotherm**	**K_f_ (mg/g)**	**n**	**R^2^**	**-**
CoFe-Ac/_P_	433.93	21.64	0.9914	-
CoFe-Ac/_Ex_	1.82	5.12	0.9848	-
CoFe-CO_3_/_A_	81.95	12.93	0.9252	-

**Table 6 materials-13-03183-t006:** Adsorption performances of azoic dyes belonging to the same family by LDHs prepared by different synthetic routes.

LDH	Ratio M^2+^/M^3+^	Dye	Initial d_003_ (Å)	Final d_003_ (Å)	Q_max_ (mg/g)	Reference
MgFe-CO_3_	3/1	Acid Brown 14	≈7.8	≈7.8	41.7	[66]
C(MgFe-CO_3_)			-	≈7.8	370.0	
MgAl-CO_3_	2/1	Congo Red	7.58	7.88	129.9	[67]
C(MgAl-CO_3_)			-	7.92	143.27	
MgAl-CO_3_	2/1	Brilliant Blue R	7.43	7.55	54.59	[68]
C(MgAl-CO_3_)			-	7.76	613.6	
MgAl-CO_3_	2/1	Direct Red 2	7.57	7,77	153.88	[25]
C(MgAl-CO_3_)			-	23.77	417.3	
MgAl-CO_3_	3/1	Acid Green 68:1	7.6	7.6	99.1	[51]
C(MgAl-CO_3_)			-	7.3	154.8	
ZnAl-CO_3_	2/1	Congo Red	7.6	Not given	Not given	[69]
C(ZnAl-CO_3_)			-	30.0	1540	
ZnAl-CO_3_		RR (X-3B)	7.6	Not given	Not given	
C(ZnAl-CO_3_)			-	7.6–8.0	390	
MgAl-SO_4_	3/1	Remazol Brilliant Red 3FB	8.12	7.9	85	[19]
ZnAl-NO_3_	2/1	Direct Red 16	8.84	11.78	69.85	[15]
MgAl-NO_3_	2/1	Reactive blue 19	8.79	8.41	281	[16]
MgAl-SDS	2/1	Direct Blue G-RB	25.5	Not given	707.76	[18]
ZnAl-Cl	2/1	Evan Blue (EB)	7.73	20.6	0.512 mmol/g	[14]
CoFe-Ac/_p_	3/1	Direct Red 2	12.70	23.77/8.28	588 (0.812 mmol/g)	This work
CoFe-Ac/_Ex_			7.67	7.67	170 (0.235 mmol/g)	
CoFe-CO_3_/_A_			7.57	7.57	127 (0.175 mmol/g)	

C: Calcined.

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
