# Peer review of "LDH-Co-Fe-Acetate: A New Efficient Sorbent for Azoic Dye Removal and Elaboration by Hydrolysis in Polyol, Characterization, Adsorption, and Anionic Exchange of Direct Red 2 as a Model Anionic Dye"

_materials, 2020, doi:10.3390/ma13143183_

Round 1

Reviewer 1 Report

The paper is quite interesting and relatively well written. The results are reliable and the conclusion is sound. I recommend its publication in Materials journal after revision according to the comments/suggestions below:

  1. The English should be revised and improved.
  2. Line 47: “They generally have high specific surfaces…” Unfortunately, the LDH materials do not have high surface areas. Only the LDH-derived materials, such as calcined (at not too high temperatures) LDH, exfoliated LDH, show high surface areas. Revise please and support the statements in this paragraph with appropriate references.
  3. Lines 51-55; lines 62-64; lines 65-69: Reference(s) needed.
  4. The authors should explain why choosing a FeCo-LDH instead of other cationic composition.
  5. Line 118: As CoFe-Ac/p sample was dried under air, the oxidation of Co(II) into Co(III) likely takes place. Therefore, why performing the exchange in inert atmosphere (line 123)?
  6. Lines 125 and 133: Specify please if drying was performed under air or in inert atmosphere.
  7. Eqns. (3) and (4) need revision: qe instead of qe and kp instead of kp, respectively.
  8. Lines 185-186: kp instead of Kp as it is not an equilibrium constant (K) but a kinetic constant (k).
  9. Line 199: “KL is the equilibrium adsorption coefficient” should be “KL is the equilibrium adsorption constant”
  10. Lines 209-212: Revise please.
  11. Eqn. (8) and line 223: kd should be KL, which is the equilibrium adsorption constant but not “the distribution coefficient”
  12. Line 235: The interlayer spacing of 12.70 Å does not allow to state “without any carbonate contamination”. The size of carbonate anion is significantly lower than that of acetate and its affinity for the interlayer space is quite high, such that it can be present in the interlayer space together with acetate. Spectroscopic analysis is necessary to rule out this hypothesis. Revise please.
  13. Figure 3 is not essential for understanding the discussion and, hence, should be moved in the supplementary material.
  14. The discussion of the IR spectra should be revised as it does not reflect the spectra presented in Figure 4.
  15. Line 280: Figure S1 shows rather a UV-Vis-NIR spectrum.
  16. Table 3 should be removed, the data being presented in text, lines 315-318.
  17. What about the textural properties of the two other samples? As they are studied as adsorbents, their textural properties are of interest and should be determined and presented in the manuscript.
  18. Line 353: “simple linear curve” should be “simple straight line”
  19. Line 359: “linear curves” should be “lines”
  20. “Whereas Hydrotalcite elaborated in aqueous medium must be calcined at 500°C in order to recover their adsorbent capacity, LDH-Ac elaborated in polyol medium manifests efficient adsorbent capacity without any further treatment.” However, while on the calcined hydrotalcite only the retention of the pollutant takes place, on the LDH-Ac the release in the effluent of the acetate ions takes place simultaneously with the retention of the pollutant! The authors should comment on this in the manuscript.

Author Response

Manuscript:  materials-877316

Dear Editor and References
Thank you for the general positive feedback on the work. Thank you also for the remarks and suggestions which enabled us to improve the manuscript both in content and in form.
The proofreading and improvement of English was carried out with the help of a colleague professor of English at the University of Paris 13 (See acknowledgments). The modifications are indicated in blue in the revised version.
Following the remarks and suggestions of the referees, some passages of the manuscript were rewritten and developed if necessary. The corresponding changes are indicated in red in the revised version.
In accordance with the advice of the publishing team, we have also reworked the writing of some parts of the manuscript, to reduce the similarities with our published articles with regard mainly to the description of the instrumentations used or data analyzes. We hope that we have succeeded in reducing the rate of these similarities.
You will find below the detailed answers for all of the referees' comments and suggestions. We hope that this revised version meets your expectations.
My best thanks

Pr. N. Jouini

Referee 1

Comments and Suggestions for Authors

The paper is quite interesting and relatively well written. The results are reliable and the conclusion is sound. I recommend its publication in Materials journal after revision according to the comments/suggestions below:

Thank you for this general positive feedback.

  1. The English should be revised and improved.

Response: We did our best to improve the quality of English helped by Mrs. Jennifer Morrice professor of English at our university who was kind enough to read and correct the manuscript. See changes indicated in blue color.

  1. Line 47: “They generally have high specific surfaces…” Unfortunately, the LDH materials do not have high surface areas. Only the LDH-derived materials, such as calcined (at not too high temperatures) LDH, exfoliated LDH, show high surface areas. Revise please and support the statements in this paragraph with appropriate references.

Response: Thank you for this remark which allows us to better specify our idea. We understood in the expression used all LDH including LDH after reconstruction.  We recognize that this is general and not very precise. Indeed, the LDHs prepared generally have small specific surfaces  less than 100 m2/g. However we can find in the literature articles citing specific surfaces greater than this value for LDH which have not undergone any calcination or exfoliation ...
We have therefore revised this passage from the manuscript to better clarify our remarks in accordance with your suggestion. Changes are indicated in red color and references were added (References 5-8).

  1. Lines 51-55; lines 62-64; lines 65-69: Reference(s) needed.

Response: As suggested, a reference was added for each paragraph: References added 9, 11, 12

  1. The authors should explain why choosing a FeCo-LDH instead of other cationic composition.

Response: Thank you for this remark which allows us to specify the general objective in which this work was carried out. Our general objective is the synthesis in polyol medium of finely divided inorganic materials having, if possible, several functions (multi-functional). It is in this context that we develop the synthesis of LDHs based on the 3d transition elements (Ni, Co, Fe). Indeed in addition to their exchange capacity, these materials open the way to other very promising applications (electrocatalysis, drug delivery, magnetic recording). LDH Co-Fe was chosen as an example from this family in this study.
A paragraph with appropriate references has been added at the end of the introduction clarifying this point (references 19, 20, 21, 22) (red color)
.

  1. Line 118: As CoFe-Ac/p sample was dried under air, the oxidation of Co(II) into Co(III) likely takes place. Therefore, why performing the exchange in inert atmosphere (line 123)?

Response: Since the first work of the group on the LHS lamellar hydroxy salts based on the transition elements 3d Ni, Co, we have shown that their synthesis in polyol medium preserves the Ni 2 +and Co

 2+ cations from their oxidation due to the reducing nature of the polyol medium (Poul et al. and Taibi et al. cited in the manuscript). We have also observed, like other authors, that once these compounds have been formed, the cations remain stable with respect to oxidation by air.
This is what we also observe here. In fact, our UV-visible analyzes carried out on LDH CoFe-Ac show that the Co 2 + cation is stable in air after several months.
We have clarified this point in the new version in the sub-paragraph on UV-Visible characters.
The exchange in an aqueous medium was carried out under an inert atmosphere by  precaution. This has been clarified in the new version

  1. Lines 125 and 133: Specify please if drying was performed under air or in inert atmosphere.

Response: Drying was performed under air. This is specified in the revised version.

  1. (3) and (4) need revision: qe instead of qe and kp instead of kp, respectively.

Response: Equations were revised

  1. Lines 185-186: kp instead of Kp as it is not an equilibrium constant (K) but a kinetic constant (k).
  2. Line 199: “KL is the equilibrium adsorption coefficient” should be “KL is the equilibrium adsorption constant”

Response: Corrections have been made

  1. Lines 209-212: Revise please.

Response: We apologize these lines do not correspond to a specific passage. However we have rewritten the paragraph (Freundlich isotherm) where these lines appear.

  1. (8) and line 223: kd should be KL, which is the equilibrium adsorption constant but not “the distribution coefficient”

        Response: Corrections have been made

  1. Line 235: The interlayer spacing of 12.70 Å does not allow to state “without any carbonate contamination”. The size of carbonate anion is significantly lower than that of acetate and its affinity for the interlayer space is quite high, such that it can be present in the interlayer space together with acetate. Spectroscopic analysis is necessary to rule out this hypothesis. Revise please.

Response: Thank you. We agree with this remark, carbonate anion can be intercalated with acetate even the distance is high. However we believe that no carbonate anion coming from dissolved CO2 in polyol is present. Indeed, it was reported that polyol molecules prevent this dissolution and avoid the formation for carbonate anion in polyol medium. This point has been addressed in the discussion paragraph. Furthermore, as shown below, the IR spectrum of CoFe-Ac/p is very similar to that reported by Choy et al. for the layered hydoxy-actetate particularly in the range of 1900-1300 cm-1 where acetate bands are present and no band characteristic of carbonate can be observed. However this does not exclude the presence of a low amount carbonate coming from air and adsorbed on the surface of the material

CoFe-Ac/P

Choy et al. 1998

In conclusion, since contamination cannot be excluded on the basis of X-ray analysis, the term “without any carbonate contamination” was suppressed. However, the IR discussion has been revised to show that the presence of carbonate must be reasonably discarded. Reference of Choy et al.1998 was added to better support our conclusion (changes are in red color).

  1. Figure 3 is not essential for understanding the discussion and, hence, should be moved in the supplementary material.

Response: We have added in this figure the SEM and TEM observations for the two other compounds and according to your suggestion this figure was moved in the supplementary material.

  1. The discussion of the IR spectra should be revised as it does not reflect the spectra presented in Figure 4.

Response: As indicated above, the IR part has been tentatively revised to justify the probable absence of carbonate in the chemical composition of the compound (changes in red color).  

  1. Line 280: Figure S1 shows rather a UV-Vis-NIR spectrum.

Response: Thank you for this remark. This was corrected.

  1. Table 3 should be removed, the data being presented in text, lines 315-318.

Response: The table was removed to supplement materials. The results concerning Cofe-Ac/ex were added.

  1. What about the textural properties of the two other samples? As they are studied as adsorbents, their textural properties are of interest and should be determined and presented in the manuscript.

Response: We apologize. In fact we carried out BET measurements on the exchanged compound Cofe-Ac / Ex. But we have not integrated them in the first version because they are similar to those of the parent compound CoFe-Ac / p. According to your suggestion, they are now added in the revised version and the description of the textural properties has been modified accordingly (changes in red color). On the other hand, unfortunately we do not have these measures for CoFe-CO3/A at present and we believe that we cannot achieve them within a reasonable time. Indeed this work is carried out by Dr. Drici Nawel mainly in Algeria with internships necessary in France to perform some measurements including the BET for the textural study. The next internship in France could only take place in several months.
It should however be noted that the results obtained here are consistent. Indeed LDH fCoFe-CO3/ A has weak adsorption capacity of the same order of magnitude as those reported in the literature for LDH containing the carbonate anion as an intercalated anion (See Table 6 and discussion).

  1. Line 353: “simple linear curve” should be “simple straight line”
  2. Line 359: “linear curves” should be “lines”

Response: changes were made

  1. “Whereas Hydrotalcite elaborated in aqueous medium must be calcined at 500°C in order to recover their adsorbent capacity, LDH-Ac elaborated in polyol medium manifests efficient adsorbent capacity without any further treatment.” However, while on the calcined hydrotalcite only the retention of the pollutant takes place, on the LDH-Ac the release in the effluent of the acetate ions takes place simultaneously with the retention of the pollutant! The authors should comment on this in the manuscript.

Response: A great thank for this remark and this suggestion which allow us to deepen and enrich the discussion. In fact, the calcination-reconstruction process significantly improves the adsorbent capacity of LDH for two main reasons: the increase in the specific surface and the intercalation of anions inside the interlayer space. However, a quick calculation shows that the intercalation rate never reaches the theoretical value fixed by the M2+/M3+. If this rate is reached, the exchange capacity should exceed 1000mg/g depending on the molar masses of the dye and the LDH. In Table 6, only the adsorption of Congo red on CLDH-Zn-Al (Ref 66) exceeds this value (1540 mg / g). In all other cases, the calcination-reconstruction process leads to average values ​​not exceeding 650 mg/g. It is therefore clear that in these cases, the compensation of the cationic layer is ensured by the carbonate ion present in the solution (coming from air) as shown by the IR studies of these systems.

So this behavior is similar to that observed in our case and discussed in the first version.
We have added this comparison in the discussion part (changes in red color).

Reviewer 2 Report

The manuscript entitled “LDH-Co-Fe-Acetate-A new efficient sorbent for azoic dye removal: Elaboration by hydrolysis in polyol, Characterization, Adsorption and anionic exchange of Direct red 2 as a model anionic dye” presents the synthesis of CoFe LDH by forced hydrolysis in polyol medium containing acetate. The exchange adsorption properties of this phases, and related ones, are studied employing Direct Red 2 anionic. Moreover, they characterized the samples in terms of PXRD, SEM, TEM, FTIR, TG-DTA.

Beside the experiments and the description of the information I do not find the manuscript to be publish in Materials-MDPI. The information that the authors present is quite valuable, however I advise the authors to reanalyse their data in order to distinguish the key points of their work.

For instance:

  1. Some important references to point out the relevance of LDHs are missing (mostly reviews for example those from O’Hare, Duan, Leroux, among others).
  2. Most of the descriptions in the introduction are not support by references (lines 46, 50, 55, 69).
  3. The authors claim “this new layered double hydroxide containing Co2+ and Fe3+ cations”. Why is this LDH new? In the last years CoFe LDHs have become a key material for electrocatalysis, hence a lot of report on CoFe LDHs are available. I do not find the novelty in this sample (in terms of composition, size, properties itself).
  4. The authors must provide an explanation of the synthetic procedure. They employed Co(2+) and Fe(2+) but finally they claimed to have Co(2+) and Fe(3+). How does the oxidation take place if “In order to avoid the oxidation of Co 2+ into Co 3+, the exchange reaction was done under inert atmosphere” (lines 122 and 123).
  5. The authors should analyse their information in terms of both compositional and structural terms as it was reported for related LDHs (J. Phys. Chem. B 2005, 109, 1, 389–393; Dalton Trans., 2014, 43, 11587-11596; Langmuir 2014, 30, 28, 8408–8415)

Author Response

Manuscript:  materials-837316

Responses to referee 2

Referee 2

The manuscript entitled “LDH-Co-Fe-Acetate-A new efficient sorbent for azoic dye removal: Elaboration by hydrolysis in polyol, Characterization, Adsorption and anionic exchange of Direct red 2 as a model anionic dye” presents the synthesis of CoFe LDH by forced hydrolysis in polyol medium containing acetate. The exchange adsorption properties of this phases, and related ones, are studied employing Direct Red 2 anionic. Moreover, they characterized the samples in terms of PXRD, SEM, TEM, FTIR, TG-DTA.

Beside the experiments and the description of the information I do not find the manuscript to be publish in Materials-MDPI. The information that the authors present is quite valuable, however I advise the authors to reanalyse their data in order to distinguish the key points of their work.

For instance:

  1. Some important references to point out the relevance of LDHs are missing (mostly reviews for example those from O’Hare, Duan, Leroux, among others).

Response: Among the great specialists in this field, we find the authors cited by the referee but also Professor Rives. It is a recent reference by this author that we cited. We also cited two other even more recent review articles in order to have an up-to-date vision in this area (Richetta et al., Mishra et al.). These references contain an exhaustive bibliography including the authors suggested by the referee. However as requested by the referee, we add a new reference (book series: Strut. Bond. 119, 2006) edited by Duan and Evans where one can find excellent review papers of these authors on LDH compounds.

  1. Most of the descriptions in the introduction are not support by references (lines 46, 50, 55, 69).

Response: In fact, the references from 2 to 4 serve as support for the development of the following paragraphs. But according to the suggestion of the referee, more precise references have been added in these paragraphs to support our purpose.

  1. The authors claim “this new layered double hydroxide containing Co2+ and Fe3+ cations”. Why is this LDH new? In the last years CoFe LDHs have become a key material for electrocatalysis, hence a lot of report on CoFe LDHs are available. I do not find the novelty in this sample (in terms of composition, size, properties itself)

Response: Thank you for this remark. We agree that LDHs based on Co-Fe are already known and in particular much sought after in recent years for their application in electrocatalysis. A paragraph has been added to clarify this point in the introduction and to show the interest of LDHs  based on the 3d transition metal (Ni, Co, Fe) as multifunctional materials in several fields (text in red color). However, these compounds are generally produced by coprecipitation and correspond to the intercalation of the carbonate ion between the layers giving a small interlayer distance. This represents a handicap for a large number of applications, in particular adsorption. In this work, we describe an LDH compound certainly based on Co and Fe but containing the acetate ion as an intercalated anion. That is why we consider it as a new compound. It was synthesized by forced hydrolysis in polyol medium a method rarely used to elaborate LDHs. It has an interlayer distance of 12.70 A. We show that this compound has significantly higher adsorption capacity than non-calcined LDH-CO3. Its adsorption is similar to those of LDH after calcination-reconstruction (memory effect).This is the interest of this material. Furthermore, its relatively more open structure can improve its electrocatalysis performances.

  1. The authors must provide an explanation of the synthetic procedure. They employed Co (2+) and Fe(2+) but finally they claimed to have Co(2+) and Fe(3+). How does the oxidation take place if In order to avoid the oxidation of Co 2+ into Co 3+, the exchange reaction was done under inert atmosphere” (lines 122 and 123).

Response: The two synthesis methods are different. 

They employed Co (2+) and Fe(2+) but finally they claimed to have Co(2+) and Fe(3+).How does the oxidation take place if

CoFe-Ac / p was prepared by forced hydrolysis in a polyol medium starting from a Co2 +  and  Fe2 + salts. Due to the operating conditions (presence of water and weakly reducing polyol power), cobalt 2+ is stable and Fe2 + is oxidized to Fe3 +. Hence the formation of LDH (Co 2 +, Fe3 +). According to this remark , the synthesis procedure was clarified along with references showing that Fe2+ can be oxidized in polyol into Fe3+ whereas Co2+ is preserved  (References Poul  et. Al.2003, Beji et al. 2005) (changes are in red color)

    “In order to avoid the oxidation of Co 2+ into Co 3+, the exchange reaction was done under inert atmosphere” (lines 122 and 123).

The second method is an exchange reaction conducted in an aqueous medium. Starting from CoFe-Ac / p (with Co2+ and Fe3+), the acetate is exchanged by the carbonate leading to the CoFe / ex LDH where the carbonate replaces the acetate between the sheets. The exchange is carried out in an inert atmosphere to avoid the possible oxidation of Co2 + to Co3 + in water.
It should be noted that Co2 + is stable with respect to oxidation in the structure of LDH as we have shown by UV analysis. This clarification was brought in the new version (spectroscopies analysis). (Changes are in red color)

  1. The authors should analyse their information in terms of both compositional and structural terms as it was reported for related LDHs (J. Phys. Chem. B 2005, 109, 1, 389–393; Dalton Trans., 2014, 43, 11587-11596; Langmuir 2014, 30, 28, 8408–8415)

Response: Thank you for advising these very interesting references. These works are in-depth structural and thermodynamic studies on the anion exchange mechanism in LDH. For this, as in the case of the work of Costantino et al. (Dalton Trans. 2014) samples taken on each isotherm at different exchanges degrees are analyzed by means of several techniques going up to the structural refinement due to the excellent crystallinity of the materials. The calculations of the thermodynamic exchange constants are also carried out. It should be noted that the systems studied lend themselves well. Indeed they are inorganic anions (Cl-, Br -...) whose characteristics are available (Oestreicher et al. Langmuir 2014). In addition, only the anion exchange is considered.
Our study concerns the adsorption of dyes on LDHs. This makes the above studies more complicated for the following reasons:
- adsorption supposes both physisorption on the surface of the particles added to the intercalation in the interlayer space by  exchange
- the material studied LDH CoFe-Ac / p has poor crystallinity which makes crystallographic studies very difficult

-Conversely to halides, dyes have variable structures and several chemical functions

 Our approach is similar to that of a large number of works on the same subject and published in renowned journals (Appl. Clay ...). Some are cited in our article including notably that of Marangoni et al. where an eminent researcher mentioned by the referee in note 1 participates.

Our study aims to specify the conditions for obtaining efficient adsorption capacity and to describe its general characteristics. Here, special attention was paid to the importance of the acetate anion on the adsorbing power of the material studied CoFe-Ac /p.  A general discussion made it possible to compare these performances among those of other LDH synthesized by other methods.
Also, thanks to X-ray diffraction, we have shown that adsorption is essentially physical in nature (surface adsorption) for low concentrations. The anionic exchange occurred at high concentrations and is accompanied by a significant increase of the interlamellar distance.
In the discussion we cite the work of Costantino et al. 2000 concerning the adsorption of fluoroscein on Zn-Al-CLO4. Indeed, great similarities exist between the two systems and you can see that their approach is very is
very close to ours.

We apologize. We are unable to respond positively to this suggestion. Indeed this supposes a total rewriting of the manuscript and above all requires experimental results which we do not have. Consequently, we have revised our manuscript in view to improving its content and form by answering all of the other comments and suggestions of the referee along with those of the other referees who gave a positive overall assessment of our approach.

Reviewer 3 Report

The paper reports the synthesis of CoFe LDH in acetate form and the study on the its ability to remove Direct red 2 from water. The work is of interest and both the characterization and discussion are satisfactory. Some observation is reported.

The Authors should justify the use of Co and Fe as metal to prepare LDH.

The formula of LDH is wrong, the charge of the lamellae is +x and not +x/n.

In the IR spectrum of CoFe-AC/p there is a small band ascribable to carbonate, perhaps a contamination of the sample occurred. LDH containing acetate anions are easily contaminate with carbonate with the exposure to the air. Have the Authors studied the stability of the CoFe-AC/p in the time?

About the morphology, TEM image shows the formation of very small particles that appear very aggregate in the SEM image. Considering the dimensions of the particles is not possible to distinguish the single particle by SEM and also it is not possible to appreciate the turbostratic disorder of the layer stacking. The discussion should be revised.

In the TG the combustion of the organic part of the sample (DEG) should be detected.

Author Response

Manuscript:  materials-877316

Dear Editor and References
Thank you for the general positive feedback on the work. Thank you also for the remarks and suggestions which enabled us to improve the manuscript both in content and in form.
The proofreading and improvement of English was carried out with the help of a colleague professor of English at the University of Paris 13 (See acknowledgments). The modifications are indicated in blue in the revised version.
Following the remarks and suggestions of the referees, some passages of the manuscript were rewritten and developed if necessary. The corresponding changes are indicated in red in the revised version.
In accordance with the advice of the publishing team, we have also reworked the writing of some parts of the manuscript, to reduce the similarities with our published articles with regard mainly to the description of the instrumentations used or data analyzes. We hope that we have succeeded in reducing the rate of these similarities.
You will find below the detailed answers for all of the referees' comments and suggestions. We hope that this revised version meets your expectations.
My best thanks

Pr. N. Jouini

Referee 2

Comments and Suggestions for Authors

The paper reports the synthesis of CoFe LDH in acetate form and the study on the its ability to remove Direct red 2 from water. The work is of interest and both the characterization and discussion are satisfactory. Some observation is reported.

We thank the reviewer for this positive overall appreciation

  • The Authors should justify the use of Co and Fe as metal to prepare LDH.

Response: Thank you for this remark also made by the reviewer 1. It allows us to specify the general objective in which this work was carried out. Our general objective is the synthesis in polyol medium of finely divided inorganic materials having, if possible, several functions (multi-functional). It is in this context that we develop the synthesis of LDH based on the 3d transition elements (Ni, Co, Fe). Indeed in addition to their exchange capacity, these materials open the way to other very promising applications (electrocatalysis, drug delivery, magnetic recording). LDH Co-Fe was chosen as an example from this

family in this study.
A paragraph with appropriate references has been added at the end of the introduction clarifying this point (references 19, 20, 21, 22) (red color).

  • The formula of LDH is wrong, the charge of the lamellae is +x and not +x/n.

Response: Sorry, the formula was corrected (red color)

  • In the IR spectrum of CoFe-AC/p there is a small band ascribable to carbonate, perhaps a contamination of the sample occurred. LDH containing acetate anions are easily contaminate with carbonate with the exposure to the air. Have the Authors studied the stability of the CoFe-AC/p in the time?

Response: Thank you for this remark.  LDH can be contaminated by carbonate when exposed to air. However, the small band in question appears at 1344 cm-1 whereas the carbonate band appears at 1364 cm-1. This small band is ascribed according to Choy et al. to delta(CH3) of the acetate anion. Furthermore, as shown below, the IR spectrum of CoFe-Ac/p is very similar to that reported by Choy et al. for the layered hydoxy-actetate particularly in the range of 1900-1300 cm-1 where acetate bands are present and no band characteristic of carbonate can be observed. However this does not exclude the presence of a low amount carbonate coming from air and adsorbed on the surface of the material.

The IR part of the manuscript has been revised to take into account the above remarks and the work of Choy et al was cited to support our analysis (red color).

CoFe-Ac/p

Ni1-x Zn2x(OH)2(CH3COO)2x.nH2O (Choy et al. 1998

             UV-visible study shows that CoFe-Ac/p is stable against oxidation of Co2+ into Co3+ when exposed to air even after exposure for several months. This point has been added in the revised version (paragraph spectroscopic characterization) (red color).  Furthermore X-Ray diffraction shows that the compound is thermodynamically stable at room temperature without variation of the interlamellar distance.

  • About the morphology, TEM image shows the formation of very small particles that appear very aggregate in the SEM image. Considering the dimensions of the particles is not possible to distinguish the single particle by SEM and also it is not possible to appreciate the turbostratic disorder of the layer stacking. The discussion should be revised.

Response: Sorry for this misinterpretation. We rewrote this part of the manuscript and added the SEM and TEM images of the other two samples (red color). According to the suggestion of reviewer 1 the corresponding figure was moved in supplement part.

  • In the TG the combustion of the organic part of the sample (DEG) should be detected.

Response: Thank you for this remark which allows us to better describe the decomposition of the compound observed by TDA/TGA. As generally reported for LDHs, the decomposition follows three steps: (i) departure of adsorbed and/or interlayer water, (ii) departure of water by dehydroxylation and (iii) finally decomposition of the intercalated anion. But for CoFe-Ac/p, the third step cannot only be explained by the decomposition of acetate. Indeed, the observed loss (28.5 %) includes that of acetate (10%), that of DEG (7%) and also the departure of oxygen due to the reduction of cobalt ions (11%).  These three phenomena occured over a wide temperature range (342-700°C) without being able to distinguish them even if inflection points are observed on the curve TG.

The discussion of the TG / TD part has been revised taking into account these precisions (red color)

Round 2

Reviewer 1 Report

The authors answered all the points raised by the reviewers and improved their work accordingly. I recommend the publication of the revised version of the manuscript as it is.

Author Response

Referee 1:

The authors answered all the points raised by the reviewers and improved their work accordingly. I recommend the publication of the revised version of the manuscript as it is.

Response: We are very pleased. Thank you very much.

Reviewer 2 Report

The author have answered all the question and revised the manuscript in detail.

I have at least two observations:

1) In the conclusions the authors claimed: "Whereas Hydrotalcite elaborated in an aqueous medium must be calcined at 500°C in order to recover their adsorbent capacity, LDH-Ac elaborated in a polyol medium manifests efficient adsorbent capacity without any further treatment."
This tense should be removed from it because it is not true. LDH containing carbonates anions need to be calcined in order to recover their adsorbent capacity, since it necesary to remove the carbonate (the most stable anion in the LDH structure). However, if LDH containing order anions (such as chloride or nitrate) they can be use as anion exchanger directely. Hence, I guess the authors need to remove this tense in order to describe the already well-known physicochemical properties in a correct way.

2) Regarding "analyse their information in terms of both compositional and structural terms", I guess the authors have a pretty nice set experiments that they can analyse in more detail for a next paper focus on how the compositional ratio can affect the disposition of molecule in the interlayer space (J. Phys. Chem. B 2005, 109, 1, 389–393

Author Response

Referee 2

The author have answered all the question and revised the manuscript in detail.

I have at least two observations:

  • In the conclusions the authors claimed: "Whereas Hydrotalcite elaborated in an aqueous medium must be calcined at 500°C in order to recover their adsorbent capacity, LDH-Ac elaborated in a polyol medium manifests efficient adsorbent capacity without any further treatment."
    This tense should be removed from it because it is not true. LDH containing carbonates anions need to be calcined in order to recover their adsorbent capacity, since it necesary to remove the carbonate (the most stable anion in the LDH structure). However, if LDH containing order anions (such as chloride or nitrate) they can be use as anion exchanger directely. Hence, I guess the authors need to remove this tense in order to describe the already well-known physicochemical properties in a correct way.

Response:

We are sorry for this confusion. In fact we meant by this passage the LDHs intercalated with the carbonate anion. In order to avoid this, we have deleted this sentence from the end of the conclusion as suggested by the reviewer and modified the end of the conclusion accordingly.

  • Regarding "analyse their information in terms of both compositional and structural terms", I guess the authors have a pretty nice set experiments that they can analyse in more detail for a next paper focus on how the compositional ratio can affect the disposition of molecule in the interlayer space ( Phys. Chem. B2005, 109, 1, 389–393) 

    Response:
We would like to warmly thank the reviewer for his suggestion to work on a new article where the emphasis will be on the mechanism of intercalation. For this, we will review the results we have and possibly perform experiments to complete them. We also thank him for taking the time to inform us of a reference from which we can draw inspiration.